# MicroRNAs in the Regulation of NADPH Oxidases in Vascular Diabetic and Ischemic Pathologies: A Case for Alternate Inhibitory Strategies?

**DOI:** 10.3390/antiox12010070

**Published:** 2022-12-29

**Authors:** Sean R. Wallace, Patrick J. Pagano, Damir Kračun

**Affiliations:** 1Pittsburgh Heart, Lung, Blood and Vascular Medicine Institute, University of Pittsburgh, Pittsburgh, PA 15261, USA; 2Department of Pharmacology and Chemical Biology, University of Pittsburgh, Pittsburgh, PA 15261, USA

**Keywords:** NADPH oxidases, NOX, miRNAs, diabetes mellitus type II, ischemia/reperfusion, obesity

## Abstract

Since their discovery in the vasculature, different NADPH oxidase (NOX) isoforms have been associated with numerous complex vascular processes such as endothelial dysfunction, vascular inflammation, arterial remodeling, and dyslipidemia. In turn, these often underlie cardiovascular and metabolic pathologies including diabetes mellitus type II, cardiomyopathy, systemic and pulmonary hypertension and atherosclerosis. Increasing attention has been directed toward miRNA involvement in type II diabetes mellitus and its cardiovascular and metabolic co-morbidities in the search for predictive and stratifying biomarkers and therapeutic targets. Owing to the challenges of generating isoform-selective NOX inhibitors (NOXi), the development of specific NOXis suitable for therapeutic purposes has been hindered. In that vein, differential regulation of specific NOX isoforms by a particular miRNA or combina-tion thereof could at some point become a reasonable approach for therapeutic targeting under some circumstances. Whereas administration of miRNAs chronically, or even acutely, to patients poses its own set of difficulties, miRNA-mediated regulation of NOXs in the vasculature is worth surveying. In this review, a distinct focus on the role of miRNAs in the regulation of NOXs was made in the context of type II diabetes mellitus and ischemic injury models.

## 1. Introduction

MicroRNAs (miRNAs) are non-coding RNAs averaging 22 nucleotides in length and are predicted to regulate the expression of 60% of human proteins [1]. In the dominant canonical pathway of miRNA biogenesis, hairpin loop-containing primary miRNAs (pri-mRNAs) transcribed from non-coding DNA are processed by the nuclear microprocessor complex into precursor miRNAs (pre-miRNAs), which are subsequently exported to the cytoplasm by an exportin 5 (XPO5)/RanGTP complex (Figure 1) [2]. Thereafter, pre-miRNA is processed by the RNase III endonuclease Dicer [3], resulting in a mature miRNA duplex wherein one strand will be utilized for messenger RNA binding [4]. Alternatively, miRNAs are produced through non-canonical pathways, such as the Dicer-independent processing of short hairpin RNA (shRNA) [5] and the microprocessor-independent maturation of short hairpin introns formed from splicing [6].

miRNAs influence protein synthesis predominantly by binding to the 3′-untranslated region (3′-UTR) of mRNAs and promoting mRNA cleavage or silencing. Mature miRNA that is fully complementary to the mRNA’s miRNA response element (MRE) on the 3′-tail of the miRNA will be sequestered for an Argonaute 2 (AGO2)-dependent endonuclease degradation [7], resulting in downregulated levels of target mRNA. However, the majority of MRE:miRNA pairs are not fully complementary, resulting in mRNA degradation by the minimal miRNA-induced silencing complex (mRISC) and resulting in target mRNA poly(A)-de-adenylation [8], de-capping [9], or exoribonuclease degradation [10], similarly decreasing target mRNA levels. On the other hand, in a number of studies, miRNAs reportedly promoted mRNA translation during cell cycle arrest [11] and under starvation conditions [12] as well as are predicted to have a transcription factor-like role [13]. However, current understanding of the prevalence of these effects and their precise mechanisms is limited.

Among multiple disease conditions increasing attention has been placed on miRNA involvement in type II diabetes mellitus and its cardiovascular and metabolic co-morbidities. In that vein, NADPH oxidases (NOXs) have been associated with type 2 diabetes mellitus and its cardiovascular and metabolic implications, such as cardiomyopathy, hypertension, atherosclerosis, various stages of liver diseases, and nephropathy [14,15,16,17].

NOXs are a family of superoxide anion-generating integral membrane oxidoreductases. Unlike other enzymes that produce reactive oxygen species (ROS) as byproducts of metabolic processes, NOXs appear to generate superoxide expressly as their primary catalytic product, transferring electrons from NADPH to molecular oxygen through prosthetic flavin and heme groups [18]. To date, four identified human NOX isozyme systems are reportedly expressed in the vasculature. These isozymes, despite having a high degree of homology, vary in structure, regulation, and roles in physiology and pathology and are defined eponymously by their core hemoprotein subunits, NOX1, NOX2, NOX4 and NOX5 [19]. In addition, there is NOX3 which is almost exclusively found in the ear, and two others which are best described in the thyroid: DUOX1 and DUOX2 [20,21]. For the function of NOX subunits (NOXs 1–4), various additional subunits are needed in different combinations/groupings: p22*^phox^*, p47*^phox^*, p67*^phox^*, p40*^phox^*, Rac1/2, NOXO1, and NOXA1 as well as maturation factors for DUOXs: DUOXA and DUOXB [15,18,19,22,23] For a general overview on the NOXs, the reader is referred to plenary reviews on the subject [15,18,22,24]. 

In the past three decades, NADPH oxidases have been extensively studied in the contexts of vascular homeostasis and cardiovascular diseases [22,25]. On the other hand, miRNAs were shown to be implicated in various aspects of cardiovascular pathologies and were presented, in essence, as biomarkers or therapeutic targets as causality was often not tested [26,27,28]. In this review, we will focus on the documented role of various miRNAs in the regulation of the NOXs’ expression and/or activity in the context of metabolic and ischemic cardiovascular pathologies.

## 2. miRNA Control in Diabetes and Diabetes-Related Cardiovascular Diseases

miRNAs have been implicated as potential diagnostic markers and/or therapeutic targets in metabolic dysregulation [29,30] and diabetes [31,32], as well as in ischemic injuries [33,34], all of which have been previously ascribed to involve NADPH oxidases [22,25]. It does not come as surprise, therefore, that the studies documenting roles of various miRNAs in the regulation of NADPH oxidases are, almost exclusively, conducted in models of type II diabetes or ischemic injury.

Along those lines, in an in vitro model of human aortic endothelial cells (HAECs) a combination of high glucose and thrombin was shown to decrease expression of miR-146a, resulting in elevated expression of target NOX4 mRNA and protein levels, and increased ROS generation [35]. Bioinformatic miR target analysis revealed a full 11-nucleotide alignment match between miR-146a and the 3′-UTR of the human NOX4 mRNA, reinforcing NOX4 as a miR-146a target [35]. Intriguingly, a luciferase reporter assay in these cells confirmed the role of miR-146a in the posttranscriptional regulation of NOX4 expression [35]. Accordingly, overexpression of the miR-146a mimic resulted in vascular protection via its anti-inflammatory effect in high glucose/thrombin-stimulated HAECs. As such, the above-described properties of miR-146a may be desirable for treating accelerated atherosclerosis and atherothrombosis in diabetes [35]. Additionally, investigating the effects of diabetes-related hyperglycemia and heightened thrombin levels, Wan et al. demonstrated significant reduction in miR-146a expression and heightened NOX4 mRNA expression levels in kidneys of streptozotocin (STZ)-induced diabetic mice compared to controls. Additionally, transfection of miR-146a into cultured immortalized proximal tubular epithelial (HK-2) cells exposed to high glucose resulted in significant decreases in protein levels of NOX4 and ROS, as well as the inflammatory markers VCAM-1 and ICAM-1 [36]. Indeed, co-transfection of a luciferase plasmid construct containing a partial NOX4 mRNA 3′-UTR including the miR-146a target site and the miR-146a mimic decreased luciferase signal to 47% of that observed with miR-control, which confirmed direct binding of miR-146a to the NOX4 3′-UTR sequence and is consistent with reduced NOX4 translation [35]. 

Hallmark clinical outcomes of diabetes are nephropathy, retinopathy, and neuropathy for which oxidative stress has been ascribed a pivotal role [37,38,39]. The pathological features of diabetic nephropathy are mesangial-cell proliferation, thickened glomerular basement membrane, glomerular hypertrophy, accumulation of ECM proteins, as well as chronic inflammation and oxidative stress caused by hyperglycemia. In a STZ-induced rat model of diabetes, Fu et al. demonstrated that mature miR-25 levels are decreased in both the renal cortex of STZ-administered rats as well as high glucose-treated cultured glomerular mesangial cells, with corresponding increases in its direct binding target NOX4 mRNA and, thusly, protein [40]. It stands to reason that STZ-induced diabetic nephropathy as confirmed by changes in urinary albumin protein and glomerular histomorphic alterations which coincided with increased NOX4 and NADPH oxidase activity. 

In vitro, high glucose, but not mannitol control, likewise induced NOX4 expression. Further inquiry revealed that a miR-25-suppressed NOX4 mRNA level likely resulted from a reduction in mRNA stability [40]. Intriguingly, among 5 miRs identified with a precise and contiguous 7 nucleotide alignment match with the 3′-UTR of NOX4 mRNA, only miR-25 was decreased in the renal cortex of rats with diabetic nephropathy presentation and in vitro high glucose treatment of mesangial cells. In mesangial cells, co-transfection of a luciferase plasmid construct chimera containing a partial NOX4 mRNA 3′-UTR (possessing the putative miR-25 binding site) along with miR-25 precursor resulted in ~40% decreased luciferase signal compared to co-transfection with control or one of the other four 3′-UTR-aligned miRs, miR-92b. However, despite the exact nucleotide match, miR-92b elicited no effect. These changes coincided with attendant changes in total NOX4 mRNA and NOX4 mRNA stability, and the effect of miR-25 was absent in cells co-transfected with NOX4 luciferase construct harboring a mutated version of miR-25 binding site [40]. Addressing these findings in the opposite manner, an antagomir for miR-25 predictably produced a stimulation in NOX4 mRNA and luciferase signal commensurate with or exceeding that of high glucose. 

Oh et al. simulated these results in a STZ mouse model discovering that mRNA levels of NOX4 and miR-25 were significantly increased and decreased, respectively, in the glomeruli of the diabetic mice compared to control littermates [41]. These changes were reproduced in vitro in mouse mesangial cells (MMCs) treated with high glucose for 72 h as compared to equimolar mannitol [41]. Both in vivo and in vitro conditions showed the levels of primary miR-25 (miR-25 precursor) increased while mature miR-25 decreased. Upon deeper exploration, the authors found that homeodomain interacting protein kinase 2 (HIPK2), responsible for maturation of miR-25, is compromised under hyperglycemic conditions leading to decreased mature miR-25 and an unabated NOX4 expression with its attendant deleterious repercussions in diabetic nephropathy [41]. Assuredly, a defined role for other players like seven in absentia homolog (SIAH) in HIPK2 activation and, hence, miR processing add complexity to maintenance of the miR-25 steady-state. This complexity lends itself well to broader exploitation of multiple targets by which to modulate miR-25 levels. Investigating diabetic nephropathy, Xu et al. identified reduced levels of miR-423-5p and elevated NOX4 mRNA as well as increased number of NOX4-positive glomerular podocytes in renal tissues obtained from diabetic patients compared to control subjects [42]. The study showed that the expression level of miR-423-5p was significantly reduced in mouse podocytes (MPC5) by high glucose treatment with simultaneously enhanced expression of NOX4 protein levels. Furthermore, in silico, NOX4 was predicted as the putative target of miR-423-5p. Dual luciferase reporter assay confirmed that the relative luciferase activity was significantly inhibited with miR-423-5p mimic and NOX4 3′-UTR WT co-transfection when compared with control group, while co-transfection of NOX4 3′-UTR mutant with miR-423-5p mimic exhibited no change in luciferase activity. Additionally, qRT-PCR and Western blot confirmed that NOX4 was regulated by miR-423-5p. The results showed that overexpression of miR-423-5p significantly inhibited the expression of NOX4 both at mRNA and protein levels. Concomitantly, the study demonstrated that cultured MPC5 treated with high glucose and transfected with miR-423-5p exhibited lower ROS and pro-inflammatory markers IL6, IL1β, MCP1, and TNFα than those treated only with high glucose [42]. A study employing an in vitro model of diabetic nephropathy documented decreased miR-485 expression in high glucose stimulated human mesangial cells (HCMs). Overexpression of miR-485 suppressed high glucose-induced ROS and proliferation of HMCs along with demonstrable increases in cyclin D1 and decreases in p21. 

The miR-485 also decreased production of pro-inflammatory cytokines such as TNFα, IL1β, and IL6 in diabetic nephropathy [43]. Wu et al. evaluated the role of miR-485 in modulating the response of HMCs to high glucose [43]. An identified NOX5 3′-UTR-binding stretch for miR-485 informed other experiments in this study and led to the discovery that miR-485 suppresses expression of NOX5, inhibits the high glucose-induced production of extracellular-matrix proteins and restrains HMC proliferation [43]. Reporter assays showed that mutation of this UTR-binding region reversed miR-485′s action on luciferase activity compared to a more than 50% reduction with the wildtype domain. 

In still another intriguing body of work, the curtailment of microRNA-25 (miR-25) maturation by high glucose or TGFβ in mesangial cells or by hyperglycemia in diabetic kidney resulted in the reversal of NOX4 gene silencing leading to increased NOX4 expression and ROS production [41].

Sensory neurons susceptibility to damage is a common complication in diabetes. Countering this damage, silent mating type information regulation 2 homolog 1 (Sirt1) has been identified as one of key gene products in neuroprotection and wound healing [44]. miRNA analysis identified miR-182 as potentially induced by Sirt1. Intriguingly then, expression of Sirt1 was downregulated in a chronic model of diabetic mouse nephropathy [44] and miR-182 was upregulated by Sirt1 overexpression in trigeminal neurons of the cornea; Sirt1 bound to the promoter of miR-182 and upregulated its transcription. Postulating it as a potential target for corneal injury [44], NOX4 was proposed as directly targeted and suppressed by miR-182 as a potential counter-regulator in diabetic nerve degeneration. In cultured primary trigeminal ganglion (TG) cells, these findings were indeed substantiated by treatment with high glucose eliciting a rise in NOX4 that was abolished with miR-182. In vivo, NOX4 protein was elevated in diabetic mouse corneal epithelia and miR-182 administration to the eye increased corneal epithelium repair [44]. Further to the point, either silencing of NOX4 or pharmacological enhancements in miR-182 (agomir) in situ produced similar protective effects on sub-basal nerves and on the corneal epithelium. In sum, these data suggest that Sirt1-stimulated miR-182 can advance corneal neuronal regeneration and nerve density by way of targeting NOX4 in diabetes. As well, they predicted that Sirt1 activators and miR-182 agomir overexpression might become treatments for diabetic keratopathy and promote corneal health.

STZ-treated type II diabetic mice that received brown fat transplantation from healthy controls displayed elevated circulating and hepatic miR-99a levels [45]. This led to significant improvement in glucose handling and non-alcoholic fatty liver disease ascribed empirically through suppression of the miR-99a target NOX4 [45]. Interestingly, diabetic patients’ circulating exosomes carry higher levels of another miR (miR-15a-3p) and have slower cutaneous wound repair attributed, in part, to inhibition of NOX5 and angiogenesis suppression [46].

## 3. Atherosclerosis

Diabetes represents a major risk factor for atherosclerosis due to hyperglycemia-induced endothelial dysfunction, inflammation, occurrence of hypercoagulability and the enhancement of atherothrombotic complications. 

In a rodent model of atherosclerosis using *apo E^−/−^* animals fed diets rich in fat and cholesterol (21% fat, 0.15% cholesterol), miR-155, reportedly possessing pro- and anti-atherogenic effects, was found to significantly reduce aortic atherosclerotic lesion area and carotid neointima formation compared to controls and those treated with miR-155 inhibitor [47]. A concomitant decrease in NOX1 associating factor (NOXA1) mRNA and protein in miR-155 mimic group were observed in addition to reduced migration of cultured mouse aortic smooth muscle cells. Migration was increased with miR-155 antagonist [47]. From these findings, the authors surmised that a lack of miR-155 would impair the vascular smooth muscle cell (VSMC) contractile phenotype and vasorelaxation by upregulating NOXA1 and, by association, the entire NOX1 oxidase and ROS levels [47]. This is indeed consistent with the work of our group showing NOX1 oxidase’s pivotal role in phenotype switching [48]. Incidentally, this shift in phenotype is germane to that occurring in atheromas. Reporter assay showed NOXA1 mRNA suppression by miR-155 [47]. Western blotting and qRT-PCR results both showed significantly decreased NOXA1 expression in miR-155 mimic group which, in contrast, was sharply increased with its antagomir/inhibitor. In vivo, aortic lesion area and carotid artery neointimal thickness in the miR-155-administered group were significantly lower compared to control and inhibitor groups. It was inferred from these findings that the function of miR-155 is highly dependent on the context and cell types (e.g., VSMC) with its early onset dynamic expression pattern and its potential suppressive effect on VSMC migration and multiple target genes, especially NOXA1. Still another study revealed that butyrate administration reportedly protects against endothelial dysfunction in *apo E^−/−^* mice fed a high fat diet (45% calories from fat) partly through reduced NOX2 expression and ROS generation. Ex vivo studies on thoracic aortas implied that this effect was mediated through butyrate-induced miR-181b upregulation by endothelial peroxisome proliferator-activated receptor (PPARδ). Ironically, while miR-181b was postulated to elicit its salutary effects via predicted NOX2 suppression, direct miR181b-NOX2 mRNA interaction was not demonstrated [49]. 

In rats fed a 2% cholesterol/0.25% cholate-enriched diet, myocardial levels of miR-25 were associated with elevated NOX4 levels and consequently oxidative/nitrative stress in the heart compared to control mice, with no detectable changes in NOX1 or NOX2 expression [50]. Similarly, Zhou et al. reported that atherosclerotic mice fed a high-fat diet exhibit depressed miR-363-3p levels and increased NOX4 levels—also observed in homocysteine-induced coronary arterial endothelial cells compared to controls [51]. Indeed, NOX4 has been identified as a direct target of miR-363-3p (luciferase reporter assay in cellulo), with miR-363-3p transfection resulting in decreased expression of the target NOX4 protein, increased cardiac aortic endothelial cell (CAECs) viability, decreased hydrogen peroxide levels, and relatedly inflammatory markers IL6, ICAM1, IL10 and IL1β [51]. Finally, PPAR activation has been implicated in the downregulation of NOX in the vasculature [52]. 

In an in vitro model of atherosclerosis, overexpression of miR-126 reduced expression of p67*^phox^* and Rac and decreased cell migration in palmitate-treated HUVECs. Thus, it appears logical to purport that miR-126 suppressed overall NOX2 (for which p67*^phox^* and Rac are essential) and, in turn, migration [53]. 

## 4. Ischemia/Reperfusion Injury Models

Herein we also include reports propounding implications for miRNAs in ischemia/reperfusion (I/R) injury models. The clinical significance of these findings is subserved by the occurrence of I/R injury in a wide array of pathological consequences of type II diabetes. I/R injury is characterized by insufficient oxygen supply as a result of restricted blood flow to tissue followed by restoration of flow. This restraint followed by a surge in flow has been documented in countless studies to cause irreversible damage to tissue. Indeed, diabetic patients suffer from increased susceptibility to myocardial and cerebral I/R injury, in good part due to increased inflammation and oxidative stress [54,55,56,57].

In a rat myocardial I/R model, left ventricular end-systolic pressure, ejection fraction, and fractional shortening were decreased, while left ventricular end-diastolic pressure was increased as was the myocardial infarction area. Concomitantly, a decrease in miR-145-5p and an increase in NOX1 were observed following I/R injury. miR-145-5p overexpression decreased superoxide anion levels. Luciferase reporter assay demonstrated that miR-145-5p binds directly to NOX1 3′-UTR. In aggregate, these data suggest that NOX1 suppression may be involved in miR-145-5p-modulated I/R injury [58].

Transfection of a microRNA-25 mimic into primary cardiomyocytes decreased superoxide anion production, while a microRNA-25 antagonist resulted in an upregulation of NOX4 protein and an increase in oxidative stress that was attenuated by the non-specific, pan-NADPH oxidase inhibitor diphenyleneiodonium [50]. These intriguing findings were substantiated by predicted targeting of the miR-25 to NOX4 and corroboration by luciferase reporter assay.

Similarly to miRNAs, long non-coding RNAs (lncRNAs) with a length greater than 200 nucleotides play key roles in development, gene programming, and gene regulation in vascular diseases [59]. lncRNA FOXF1-AS1 adjacent non-coding developmental regulatory RNA (FENDRR) is found to be a critical gene in the development of vascular or malignant diseases [60]. In patients with essential hypertension, a comorbidity for type II diabetes, the competitive endogenous RNA (ceRNA) FENDRR was found to be elevated, while free circulating miR-423-5p was lower. Indeed, FENDRR has been proposed as an endogenous antagonist of miR-423-5p by modeling in support of direct binding of the two non-coding RNA subtypes. In vitro studies utilizing HUVECs reported that miR-483-5p is putatively bound and neutralized by FENDRR, which, in turn, causes a permissive rise in NOX4 by disinhibition. miR-483-5p upregulation resulted in decreased levels of NOX4 protein through direct mRNA interference as shown by NOX4 mRNA-luciferase activity and Western blot [61]. The consequence of an increased FENDRR and attendant decrease in miR-483-5p in essential hypertension appears tantamount to a rise in NOX4 and protection against endothelial cell proliferation and migration.

Similar to myocardial I/R, both acute hyperglycemia and chronic diabetes have been demonstrated to exacerbate I/R brain damage. Moreover, diabetes accelerates the development of neuronal damage, increases infarction volume, and induces post-ischemic seizures under diverse mechanisms [62,63,64]. In the cerebral I/R rat model increased miR-125b was shown to be accompanied by decreased serine-threonine-selective casein kinase 2 (CK2). In vitro, activation of NOX2 and NOX4 and neuronal (PC-12 cells) deficit triggered by oxygen/glucose deprivation (OGD/R) and reperfusion were reversed by the inhibition of miR-125b and a rise in the NOX-suppressive action of CK2 [65]. In yet another study, NOX2 was upregulated and miR-126a-5p was downregulated in the brains of I/R rats. miR-126a-5p mimic reduced the neurological deficit, infarct volume, brain fluid content, oxidative stress, and apoptosis in I/R rats [66]. Importantly, miR-126a-5p targets NOX2 directly and negatively regulates the oxidase. Moreover, miR-126a-5p mimic enhanced viability and inhibited oxidative stress and apoptosis in OGD/R-treated SH-SY5Y (human neuroblast-like) cells, while miR-126a-5p inhibitor expectedly had the opposite effect [66]. Furthermore, NOX2 overexpression antagonized the protective effects of miR-126a-5p mimic on OGD/R-induced cell injury [66]. Similarly, employing I/R injury rat model and a SHSY5Y cell culture OGD/R model, increased miR-454 levels were found to alleviate I/R-induced NOX4 upregulation and brain damage, suggesting that miR-454 might be vital to protect neurons and reduce brain damage by regulating NOX4. This implies that the miR could act as a neuroprotective agent in ischemic stroke [67].

Another report on rats subjected to I/R identified upregulation of NOX2 and concomitant downregulation of miR-532-3p. In addition, the plasma expression of miR-532-3p was also significantly decreased. It is tempting to speculate, therefore, that there could be a negative feedback loop at work during the early stages of I/R that puts the “brake” on rampant NOX2 expression which is released as I/R ensues. A reporter gene assay with an expression vector containing 3′-UTR of NOX2 (WT or MU-mutated) conducted with miR-532-3p mimics revealed a significant decrease in the relative luciferase activity in cells transfected with the NOX2-WT plasmid, but not with NOX2-MU. As for the others, this rather convincingly drives home the notion that miR-532-3p suppresses NOX2 expression by directly binding to its 3′-UTR [68].

A comparable study on cerebral I/R rat model identified in silico potential miRNAs targeting NOX2 to be: miR-93, miR-182, miR-92a, miR-134 and miR-652 [69]. After evaluating their expression levels in brain tissues, only the miR-652 was found to be significantly decreased in brain tissues subjected to I/R injury, while the others were all increased. These data suggest that low expression of miR-652 might contribute to higher expression of NOX2 and result in the ROS rebound and overshoot seen in I/R cerebral injury. To corroborate these data, expression plasmids containing either the wild type 3′-UTR of NOX2 or mutated 3′-UTR of NOX2 were co-transfected with miR-652 mimics into SH-SY5Y cell line. miR-652 mimic significantly decreased the relative luciferase activity in the NOX2-WT group but not in the NOX2-MU group, authenticating that miR-652 regulated the expression of NOX2 by binding to its 3′-UTR [69]. In a study employing the transient middle cerebral artery occlusion (tMCAO) rat model, the microRNA profile targeting NOX2 and NOX4 genes was analyzed as was its response to NOX2/4 inhibitor VAS2870 [70]. Using in silico analysis, miRNA microarray and confirmation by qRT-PCR, the authors found the expressions of miR-29a-5p, miR-29c-3p, miR-126a-5p, miR-132-3p, miR-136-3p, miR-138-5p, miR-139-5p, miR-153-5p, miR-337-3p, and miR-376a-5p were significantly downregulated in the ischemic hemisphere in rats with tMCAO compared with sham-operated rats. However, only the expression of miR-29a-5p, miR-29c-3p, miR-126a-5p, and miR-132-3p were found to be downregulated after mechanical reperfusion and restored by NOX inhibitor VAS2870 treatment. This suggests that the upregulation of these four miRNAs might contribute to a protective role of VAS2870 against cerebral ischemic reperfusion injury and hemorrhagic transformation [70].

In a study using a transgenic APP/PS1 mouse model of Alzheimer’s disease, TargetScan^TM^ analysis identified a potential binding site for miR-204-3p in the 3′-UTR of NOX4 [71]. Indeed, relative luciferase activity was significantly reduced when mouse neuroblastoma (N2a) cells were co-transfected with miR-204-3p and pGL3-NOX4-3′-UTR, but not with the mutant NOX4 luciferase construct. Unlike demonstrably unchanged mRNA levels, NOX4 protein levels were reduced in the hippocampus of Lv-miR-204-treated APP/PS1 mice. Moreover, the protein level of NOX4 was increased in the hippocampus of control APP/PS1 mice untreated with miR-204-3p, substantiating that NOX4 protein expression is a target of miR-204-3p [71].

## 5. Conclusions and Perspectives

In pursuit of our understanding of miRNA regulation of NADPH oxidases to date (summarized in Table 1, Table 2 and Table 3), the majority of reports focus on NOX4, likely due to its broadly appreciated activation via transcription and translation. The increasing focus should also be given to the independent unit NOX5, as well as the multiple membranal and cytosolic components of the NOX1 and NOX2 complexes in cardio-metabolic-related pathogenesis. Indeed, direct miRNA regulation of NOX1 has been illustrated previously in models of myocardial reperfusion injury [58], while NOX2 regulation has been shown in ischemic stroke (Figure 2) [66].

ROS derived from NADPH oxidases play an important role in metabolic disorders (e.g., diabetes mellitus type II and metabolic syndrome) and associated cardiovascular diseases including endothelial dysfunction, hypertension, hypercholesterolemia, coronary heart disease, heart failure and stroke [72,73,74,75]. The progression of metabolic dysfunction from nutrient excess to obesity and type II diabetes appears to be related to the supersedence of low-level, beneficial NOX signaling for exaggerated, deleterious expression of specific NOX isoforms and thus ROS production. In metabolic disease, excessive ROS production from NOX enzymes contributes to inflammation, increased vascular tone, insulin resistance, and the progression of atherosclerosis. As alluded to above and with NOX isozymes displaying fundamental roles in physiology, it is likely that pan inhibition of NOX enzymes will not be a plausible therapeutic strategy. Therefore, targeting specific NOX isoforms potentially in a tissue- or cell-type-controlled manner is expected to provide a more fine-tuned therapeutic strategy in addressing metabolic disease pathologies. Perhaps more importantly, the large body of data surveyed in this review put forward the notion that miRNAs serve critical counter-regulatory roles in disease. The manner in which any one or more miR or miR family performs this role is expected to (a) be far more complex than the singular effects described; and (b) involve factors yet unknown that orchestrate the effect of multiple miRs. Moreover, usage of miRNAs even in vitro, may come at the expense of off-target effects and, indeed, even unwanted effects on a particular NOX isoform under certain conditions, with preclinical application being far more complex taking into account complicated drug design, delivery and efficacy. Admittedly, the relative promiscuity of miRNAs toward an array of proteins’ translation, rather than a singular target in this context, render their therapeutic use limited at this time and may require an in-depth inquiry into the entire cluster of proteins suppressed in the context of disease and wellbeing.

Given the NOX family’s active roles in hypertension, diabetic atherosclerosis, and endothelial dysfunction, increasing emphasis has been placed on understanding possible pharmacological interventions. One challenging but worthwhile goal for NOX therapeutics is isozyme selectivity, which is being vigorously pursued by a number of groups including our own [76,77,78,79,80,81]. In the meanwhile, miRNA targeting of specific NOX proteins at the translational level is a promising new avenue for consideration. In the latter case, an alternative to typical enzyme-targeted interventions is achieved through destabilizing and degrading mRNA constructs, either by administering miRNA mimics to downregulate NOX expression or administering antagomirs to bind and neutralize inhibitory miRNAs which consequently enhance NOX expression. While no miRNA-related drugs are currently available for FDA-approved use, multiple Phase I and Phase II clinical trials studying multiple diverse diseases are being performed utilizing antagomirs that neutralize complementary miRNAs.

Unquestionably, more studies are needed to investigate the roles of other vascular NOXs and their miRNA counter-regulators in the progression of obesity-related vascular diseases, both to better elucidate the NOXs’ pathological effects and to engineer RNA, peptidic and small-molecule druggable therapeutics.

**Table 3 antioxidants-12-00070-t003:** Differential Expression of miRNAs and NOX Isoforms in Models of Ischemia/Reperfusion Injury Legend: SD—Sprague Dawley; MI/R—Myocardial Ischemia/Reperfusion; LAD—Left Anterior Descending Coronary Artery; CI/R—Cerebral Ischemia/Reperfusion; ICA—Internal Carotid Artery; PCA—Posterior Cerebral Artery; MCA—Middle Cerebral Artery; ACA—Anterior Cerebral Artery; AD—Alzheimer’s Disease; APP—Amyloid Precursor Protein; PS1—Human Presenilin 1.

Pathology	Model	miRNA Identified	NOX Regulation	Ref.
MI/R	SD rats LAD artery ligation for 30 min/reperfusion for 3 or 24 h	↓ miR-145-5p	↑ NOX1	[58]
CI/R	SD rats occlusion of ICA, PCA, MCA, ACA for 2h/reperfusion for 24 h	↓ miR-126a-5p	↑ NOX2	[66]
CI/R	SD rats occlusion of ICA, PCA, MCA, ACA for 2 h/reperfusion for 24 h	↓ miR-454	↑ NOX4	[67]
CI/R	SD rats occlusion of ICA, PCA, MCA, ACA for 2 h/reperfusion for 24 h	↓ miR-652	↑ NOX2	[69]
CI/R	SD rats occlusion of MCA for 2 h/reperfusion for 24 h	↓ miR-92b-3p	↑ NOX4	[82]
CI/R	SD rats occlusion of ICA for 2 h/reperfusion for 24h	↓ miR-523-3p	↑ NOX2	[68]
AD	APPswe/PS1dE9 mice APP/PS1 mice, 6 mos old	↓ miR-204-3p	↑ NOX4	[71]

## Figures and Tables

**Figure 1 antioxidants-12-00070-f001:**
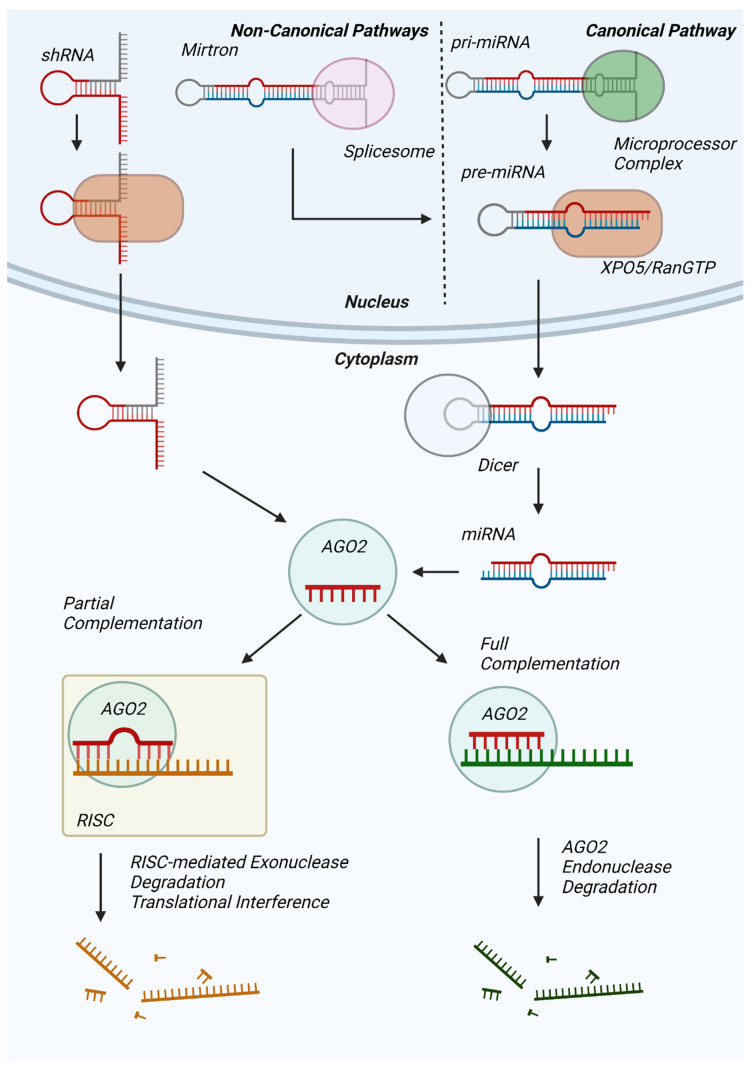
Overview of miRNA synthesis, maturation, and mRNA degradation. Pri-miRNAs transcribed by RNA Polymerase II (canonical) or Mirtrons produced by the spliceosome (non-canonical) are processed by the Drosha/DGCR8 microprocessor complex into pre-miRNAs for subsequent XPO5/RanGTP complex-mediated export. Cytosolic Dicer further processes pre-miRNA into mature miRNA for subsequent Ago2 binding. shRNA (non-canonical) can be exported from the nucleus via XPO5-RanGTP and be incorporated directly into Ago2. The Ago2-miRNA complex targets mRNA for degradation either via Ago2 endonuclease activity (full complementarity) or recruitment of additional proteins for RISC-mediated mRNA degradation or translational interference.

**Figure 2 antioxidants-12-00070-f002:**
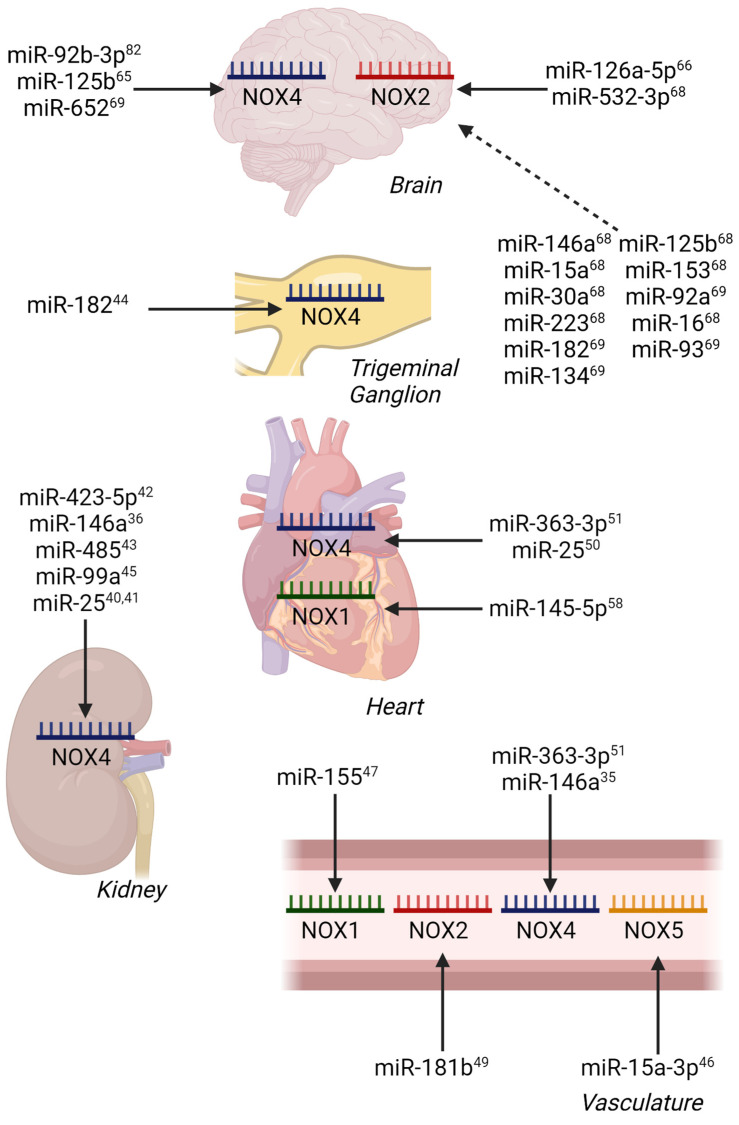
miRNA Regulation of NOX Isoforms in Models of Vascular, Metabolic, and Ischemia/Reperfusion Injury by Tissue. Confirmed miRNA-mRNA interactions are indicated by solid arrows, while predicted miRNA-mRNA interactions are indicated by dashed arrows. Superscript numbers indicate associated references.

**Table 1 antioxidants-12-00070-t001:** Differential Expression of miRNAs and NOX Isoforms in Models or Patients with Obesity or Comorbid Diseases Legend: HFD—High Fat Diet; CHL—Cholesterol; T1DM—Type I Diabetes Mellitus; STZ—Streptozotocin; QD—Daily; DM—Diabetes Mellitus; FG—Fasting Blood Glucose; T2DM—Type 2 Diabetes Mellitus; Vit—Vitamin; PTU—Propylthiouracil; Inj—Injection.

Pathology	Model/Patients	miRNA Identified	NOX Regulation	Ref.
Atherosclerosis	*apoE^−/−^* Mouse, HFD (21% fat, 0.15% CHL), 12 wks; miR-155 administered	↑ miR-155	↓ NOXA1	[47]
Atherosclerosis	*apoE^−/−^* Mouse HFD (45% fat; 0.5 mg/g butyrate), 10 wks	↑ miR-181b	↓ NOX2	[49]
T1DM-Nephropathy	Rat, STZ (50 mg/kg QD), 5 d; DM = FG ≥ 15 mM	↓ miR-25	↑ NOX4	[40]
T1DM-Nephropathy	Mouse, STZ (50 mg/kg QD), 5 d DM = FG > 16.65 mM	↓ miR-25	↑ NOX4	[41]
T1DM-Nephropathy	Mouse, STZ (55 mg/kg QD), 5 d	↓ miR-146a	↑ NOX4	[36]
T2DM-Retinopathy	BKS.Cg-m+/+Leprdb/J (db/db) Mouse DM = FG > 15 mM	↓ miR-182	↑ NOX4	[44]
T2DM	Mouse, HFD (60% fat, 20% carbohydrates), 8 wks STZ (120 mg/kg x1 at 4wks) DM = FG ≥ 13.9 mM. Brown fat transplantation.	↑ miR-99a	↓ NOX4	[45]
Dyslipidemia	Rat, HFD (2% CHL, 0.25% cholate), 12 wks	↓ miR-25	↑ NOX4	[50]
Atherosclerosis	Mouse, HFD (1.25 g/kg Vit D3, 2% CHL, 0.5% cholate, 3% lard, 0.2% PTU), 3 mos; 3 g/kg Vit D3 Inj. 1x/mos	↓ miR-363-3p	↑ NOX4	[51]
T2DM	Patients aged 45–60 y with foot wounds Circulating Serum Exosomes harvested	↑ miR-15a-3p	↓ NOX5	[46]
T2DM-Nephropathy	Adult T2DM patients Renal Fibroblasts harvested	↓ miR-423-5p	↑ NOX4	[42]

**Table 2 antioxidants-12-00070-t002:** Differential Expression of miRNAs and NOX Isoforms in vitro Legend: MPC5s—Immortalized Mouse Podocyte Line; T2DM—Type II Diabetes Mellitus; NG—Normal Glucose; HG—High Glucose; HAEC—Human Aortic Endothelial Cells; HMCs—Human Mesangial Cells; CAECs—Coronary Arterial Endothelial Cells; SH-SY5Y—Neuroblastoma cell line; OGD/R—Oxygen and Glucose Deprivation/Reoxygenation; H/R—Hypoxia/Reoxygenation.

Cell Line	Model/Patients	miRNA Identified	NOX Regulation	Ref.
MPC5	NG vs HG (5 mM/30 mM D-glucose) miR-423-5p administered	↑ miR-423-5p	↓ NOX4	[42]
HAECs	NG vs HG (25 mM L-glucose/25 mM D-glucose) + Thrombin (2 U/mL)	↓ miR-146a	↑ NOX4	[35]
HMCs	NG vs NG + Mannitol vs HG (5.5 mM D-glucose/5.5 mM D-glucose + 24.5 mM Mannitol/30 mM D-glucose)	↓ miR-485	↑ NOX5	[43]
CAECs	Homocysteine (0.25 mM), 24 h	↓ miR-363-3p	↑ NOX4	[51]
SH-SY5Y	SH-SY5YOGD/R Hypoxia (N2/CO2 95:5), 10 min Hypoxia + Glucose-free medium, 2 h Normoxia and NG, 24 h	↓ miR-126a-5p	↑ NOX2	[66]
SH-SY5Y	OGD/R Hypoxia, 10 min Hypoxia + Glucose-free medium, 2 h Normoxia and NG, 24 h	↓ miR-454	↑ NOX4	[67]
SH-SY5Y	H/R Hypoxia (N_2_/CO_2_ 95:5) + PBS, 5 h Normoxia, 20 h	↓ miR-652	↑ NOX2	[69]

## Data Availability

Not applicable.

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
