# Peer review of "MicroRNAs in the Regulation of NADPH Oxidases in Vascular Diabetic and Ischemic Pathologies: A Case for Alternate Inhibitory Strategies?"

_antioxidants, 2022, doi:10.3390/antiox12010070_

Round 1

Reviewer 1 Report

In this review, the authors discuss the documented role of miRNAs in the regulation of the NOXs’ expression and/or activity in experimental models of type II diabetes and ischemic cardiovascular pathologies. Although the majority of reports examined by the authors focus on NOX4, likely due to its activation via transcription and translation, this review is exhaustive and well-organized.

1. This review focus on the role of various miRNAs in the regulation of the NOXs’ expression and/or activity, in the context of metabolic and ischemic cardiovascular pathologies. Although administration of miRNAs is not allowed with current metodologies, the knowledge of miRNA regulation of NOXs in type II diabetes mellitus and ischemic injury models is of great interest.

2. The topic is original since the informations reported in the review address the specific field of the role of miRNA regulation of NOXs in metabolic (type II diabetes mellitus), ischemic and cardiovascular diseases.

3. In most of the other published papers, miRNA regulation of NOXs is addressed primarily in cancer. A limited number of articles describe this effect in metabolic and cardiovascular diseases. This review summarizes exhaustively the data reported in the literature.

4. The article of Wallace et al., is a review. Therefore, no specific improvements of methodology neither experimental controls are required. 5. The authors conclude their review with a section entitled “Perspectives” where they describe appropriately the possible therapeutical interventions of miRNA regulation of NOXs.

6. The references are appropriate.

7. The 2 Figures and the 3 Tables are clear and perfectly legible.

I suggest the publication in the present form.

Reviewer 2 Report

Wallace et al. present a review article on the concept of selective inhibition of NADPH oxidase by miRNA. The authors focus their review on type II diabetes mellitus and ischemic injury models. The idea of selectivity of Noxs by MiRNAs seems surprising since MiRNAs are in essence molecules controlling the expression of many targets at the same time. This notion of selectivity is therefore complicated to understand in the context of this review.

The authors refer in three tables to the MiRNAs impacting the Noxs in the context of targeted pathologies. There are thus between 20 and 30 MiRNAs involved in the regulation of Noxes.

-It would be interesting to describe for each of these MiRNAs the diversity of downstream targets in a specific table. This would shed more light on what the authors mean by selective Nox inhibition. And describe in the text for each MiRNA was are the other target common to Nox pathways if there is some.

- Tables 1-3 need to be improved in their presentation (centered text etc..).

-There are still some typo's with bold references and some without.
